# Clinical Surveillance vs. Anticoagulation For low-risk patiEnts with isolated SubSegmental Pulmonary Embolism: protocol for a multicentre randomised placebo-controlled non-inferiority trial (SAFE-SSPE)

Christine Baumgartner [1], Frederikus A Klok,[2] Marc Carrier,[3] Andreas Limacher,[4] Jeanne Moor,[1] Marc Righini,[5] Jürg-Hans Beer,[6] Martina Peluso,[1] Damiana Rakovic,[1] Menno V Huisman,[2] Drahomir Aujesky[1]

For numbered affiliations see end of article.

**Correspondence to**
Dr Christine Baumgartner;
Christine.Baumgartner@insel.ch

## ABSTRACT

**Introduction** The clinical significance of subsegmental pulmonary embolism (SSPE) is currently unclear. Although growing evidence from observational studies suggests that withholding anticoagulant treatment may be a safe option in selected patients with isolated SSPE, most patients with this condition receive anticoagulant treatment, which is associated with a 90-day risk of recurrent venous thromboembolism (VTE) of 0.8% and major bleeding of up to 5%. Given the ongoing controversy concerning the risk-benefit ratio of anticoagulation for isolated SSPE and the lack of evidence from randomised-controlled studies, the aim of this clinical trial is to evaluate the efficacy and safety of clinical surveillance without anticoagulation in low-risk patients with isolated SSPE.

**Methods and analysis** SAFE-SSPE (Surveillance vs. Anticoagulation For low-risk patiEnts with isolated SubSegmental Pulmonary Embolism, a multicentre randomised placebo-controlled non-inferiority trial) is an international, multicentre, placebo-controlled, double-blind, parallel-group non-inferiority trial conducted in Switzerland, the Netherlands and Canada. Low-risk patients with isolated SSPE are randomised to receive clinical surveillance with either placebo (no anticoagulation) or anticoagulant treatment with rivaroxaban. All patients undergo bilateral whole-leg compression ultrasonography to exclude concomitant deep vein thrombosis before enrolment. Patients are followed for 90 days. The primary outcome is symptomatic recurrent VTE (efficacy). The secondary outcomes include clinically significant bleeding and all-cause mortality (safety). The ancillary outcomes are health-related quality of life, functional status and medical resource utilisation.

**Ethics and dissemination** The local ethics committees in Switzerland have approved this protocol. Submission to the Ethical Committees in the Netherlands and Canada is underway. The results of this trial will be published in a peer-reviewed journal.

**Trial registration number** NCT04263038.

<div style="border:1px solid #000">

**Strengths and limitations of this study**

► This is the first randomised trial comparing the safety and efficacy of a management strategy without anticoagulation and anticoagulant treatment in low-risk patients with isolated subsegmental pulmonary embolism (SSPE).

► We chose patient-centred outcomes and economically relevant efficiency measures that are relevant to both patient and healthcare professionals.

► As patients with isolated SSPE and a high risk of adverse events (eg, patients with cancer) are excluded from this trial due to safety reasons, generalisability of the results will be limited to low-risk patients with isolated SSPE.

</div>

## INTRODUCTION

Depending on thromboembolic burden and patient factors, the clinical spectrum of pulmonary embolism (PE) ranges from asymptomatic cases to massive PE with haemodynamic collapse.[1 2] Anticoagulant treatment for at least 3 months effectively reduces the risk of recurrent venous thromboembolism (VTE).[3 4] The benefit of anticoagulation, however, comes at the cost of potentially disabling and life-threatening bleeding events, with a 90-day risk of major bleeding of up to 5%.[5–8]

The widespread introduction and technological advances of multi-detector CT pulmonary angiography (CTPA) have led to an 80% increase in the diagnosis of acute PE between 1998 and 2006.[9] This increase is in part due to an increase in the detection of small, peripheral PE limited to the subsegmental pulmonary arteries, that is, subsegmental PE

(SSPE),[10 11] which currently comprise 10%–15% of cases with PE.[12–15] However, the clinical significance of isolated SSPE is questionable. Epidemiological evidence suggests overdiagnosis: despite the increase in PE diagnoses in recent years, PE-related mortality has remained stable or has even decreased.[9 16] The positive predictive value of CTPA to diagnose SSPE is as low as 25% (compared with a composite reference standard with ventilation-perfusion lung scanning with or without lower limb venous ultrasonography, or pulmonary digital-subtraction angiography),[17] and interobserver agreement between radiologists for SSPE diagnosis is only fair,[18] indicating that differentiation between true emboli and artefacts in the subsegmental pulmonary arteries is difficult. Small PE may even occur in healthy individuals without clinical consequences,[19–21] suggesting that SSPE may be the result of the physiological filter function of the lung to protect the systemic circulation.

Whether patients with isolated SSPE benefit from anticoagulant treatment is currently uncertain.[4] There is growing evidence from observational studies that withholding anticoagulation may be safe in patients with isolated SSPE who are at low risk of recurrent or progressive VTE,[22–27] but most of such patients currently receive anticoagulant treatment,[28–31] potentially exposing them to an unnecessary risk of bleeding. Given the ongoing controversy and clinical equipoise about the risk-benefit ratio of anticoagulation for isolated SSPE, we aim to evaluate the efficacy and safety of clinical surveillance without anticoagulation compared with standard anticoagulation treatment in low-risk patients with SSPE in a randomised clinical trial.

## METHODS AND ANALYSIS

This study protocol has been developed according to the Standard Protocol Items: Recommendations for Interventional Trials guidelines.[32]

### Objectives and hypotheses

The primary objective of this randomised trial is to compare the frequency of symptomatic, recurrent VTE in low-risk patients with isolated SSPE randomised to receive clinical surveillance plus placebo or clinical surveillance plus anticoagulant treatment with rivaroxaban. As a secondary objective, the frequency of clinically significant bleeding and all-cause mortality is compared in the two groups. Ancillary endpoints include health-related quality of life, functional status, and medical resource utilisation. We hypothesise that clinical surveillance without anticoagulant treatment is non-inferior to anticoagulation in terms of recurrent VTE, while resulting in fewer clinically significant bleeding events and similar all-cause mortality. We also hypothesise that clinical surveillance, compared with anticoagulation, improves health-related quality of life and functional status and reduces medical resource utilisation.

### Study design and setting

SAFE-SSPE (clinical Surveillance vs. Anticoagulation For low-risk patiEnts with isolated SubSegmental Pulmonary Embolism) is an investigator-initiated, multicentre, randomised, placebo-controlled, double-blind, parallel-group non-inferiority trial. Patients with isolated SSPE at low risk for VTE progression or recurrence and without concomitant deep vein thrombosis (DVT) are randomly assigned in a 1:1 ratio to receive clinical surveillance plus placebo or clinical surveillance plus anticoagulation with rivaroxaban (figure 1). Randomisation is blocked and stratified by study site. To ensure concealment of allocation, the allocation sequence is generated by a data manager not involved in the study, using a computer-generated randomisation schedule. All participants, care providers, investigators, study personnel, members of the outcomes adjudication committee, data management personnel and analysts are blinded to group assignment. Eligible patients are recruited in at least 27 university and medium-volume to high-volume non-university teaching hospitals in Switzerland, the Netherlands and Canada.

### Selection of patients

Consecutive patients aged ≥18 years with objectively diagnosed symptomatic or asymptomatic isolated SSPE on CTPA based on the assessment of the local radiologist at the time of patient presentation are eligible for study participation after provision of informed consent (table 1). Isolated SSPE is defined as multi-detector CTPA demonstrating an intraluminal filling defect in ≥1 subsegmental pulmonary artery (4th order or higher) without filling defects visualised at more proximal pulmonary artery levels.[24 33 34] Isolated subsegmental defects are classified as either single (one subsegmental vessel involved) or multiple (≥2 subsegmental vessels involved). Patients with both symptomatic and incidentally detected, asymptomatic isolated SSPE are potentially eligible because symptoms of PE may be subtle and difficult to elucidate, and symptomatic and asymptomatic PE appear to have a similar prognosis.[35] We will enrol both patients with single and multiple isolated SSPE because there is no convincing evidence that these conditions are prognostically different.

The exclusion criteria were selected based on a high early risk of VTE recurrence (≥8%), the presence of cardiopulmonary compromise (ie, hypotension or hypoxaemia), contraindications to anticoagulant treatment with rivaroxaban, and a high risk of confounding (table 2).[4 14 36–44] All exclusion criteria are listed in table 2. Because PE usually originates from a thrombus in the deep leg veins,[45] potentially eligible patients systematically receive a single bilateral whole-leg compression ultrasonography (CUS) examination to exclude concomitant DVT, which is a well-known prognostic factor for mortality in patients with PE and represents an absolute indication for therapeutic anticoagulation.[46] If a proximal or distal DVT (with an incompressible distal vein diameter of ≥5 mm)[47] is detected, patients are excluded from

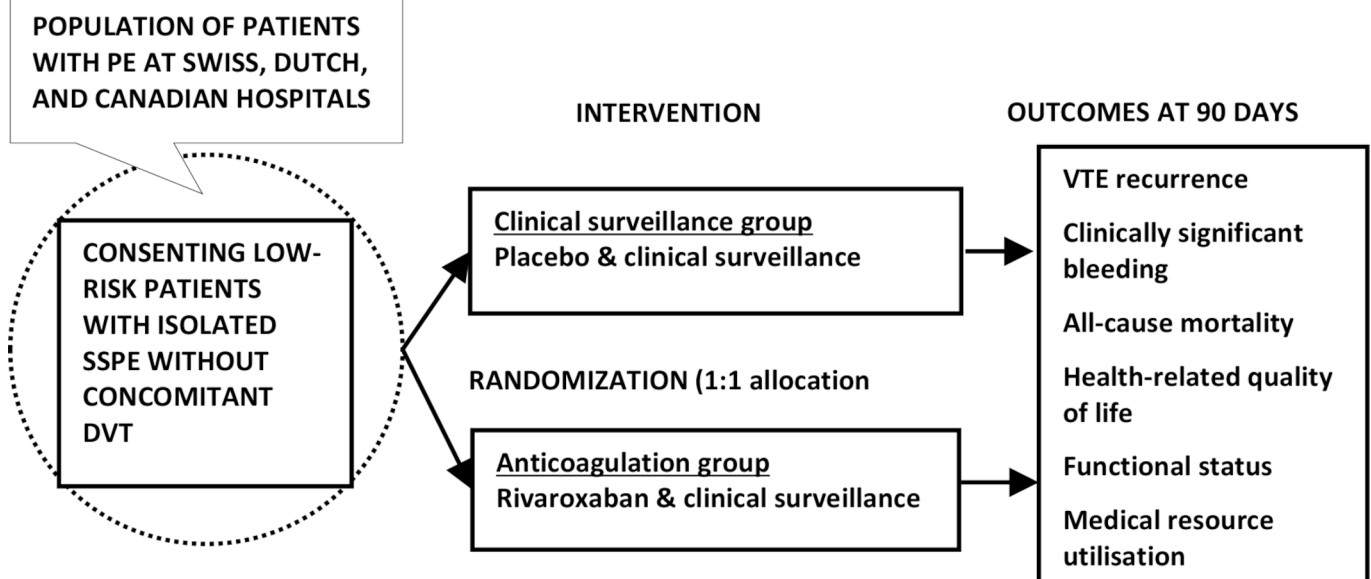

**Figure 1** Overview of the study design. SAFE-SSPE is a randomised, placebo-controlled, parallel-group non-inferiority trial. Low-risk patients with isolated SSPE without concomitant DVT are randomly assigned in a 1:1 ratio to receive placebo ('clinical surveillance group') or anticoagulant treatment with rivaroxaban ('anticoagulation group'). The primary study outcomes are symptomatic recurrent VTE within 90 days of randomisation. Secondary outcomes include clinically significant bleeding and all-cause mortality, and ancillary outcomes will be health-related quality of life, functional status and medical resource utilisation. DVT, deep vein thrombosis; PE, pulmonary embolism; SSPE, subsegmental pulmonary embolism; VTE, venous thromboembolism.

study participation and treated at the discretion of their managing physician. We cannot fully exclude the small possibility that an eligible patient with SSPE in whom the presence of a leg vein DVT was ruled out by a bilateral whole-leg CUS may have a concomitant isolated iliac vein thrombosis or a thrombosis at an unusual site (eg, in the cerebral, splanchnic or ovarian veins). However, these are rare thrombotic conditions that often occur in specific clinical situations representing study exclusion criteria (cancer, pregnancy, puerperium).[48] The risk that such patients would present with an isolated SSPE is even lower. Although whole-leg CUS is not an effective method to exclude an isolated iliac vein thrombosis, a meta-analysis of randomised trials and prospective management studies

| Table 1 | Study population, intervention, control and outcomes |
|---|---|
| Population | Consecutive adult low-risk patients with an objective diagnosis of *isolated subsegmental PE* who have no concomitant DVT. |
| Intervention | *Clinical surveillance plus a matching rivaroxaban placebo*, one tablet two times per day for the first 21 days, followed by one tablet once daily for the remaining 90-day study period. Clinical surveillance is done at 10, 30 and 90 days following randomisation by phone or by in-person visits, depending on local practice. At each contact, trained study personnel complete an assessment of symptoms and review for suspected recurrent VTE and bleeding using a checklist of predefined questions. Patients are also instructed to contact study personnel or report to the ED immediately if any symptoms/signs compatible with recurrent VTE or significant bleeding occur. |
| Control | *Clinical surveillance plus anticoagulation with rivaroxaban*, dosed at 15 mg two times per day for the first 21 days, followed by 20 mg once daily for the remaining 90-day study period. The same surveillance schedule as in the clinical surveillance group is used. Patients also receive the same instructions to contact study personnel or report to the ED if any signs or symptoms of VTE or significant bleeding occur. |
| Outcomes | Primary outcome: *recurrent, clinically symptomatic, objectively confirmed VTE* within 90 days of randomisation, defined as recurrent fatal or non-fatal PE or lower limb DVT (efficacy). Secondary outcomes: *clinically significant bleeding and all-cause mortality* at 90 days of randomisation (safety). Ancillary outcomes: *health-related quality of life, functional status and medical resource utilisation* at 90 days of randomisation. In a post-hoc analysis, *radiological interobserver agreement* for SSPE will be assessed. |

DVT, deep vein thrombosis; ED, emergency department; PE, pulmonary embolism; SSPE, subsegmental pulmonary embolism; VTE, venous thromboembolism.

**Table 2** Exclusion criteria and their rationale

| Exclusion criterion | Rationale |
|---|---|
| Presence of leg DVT or upper extremity DVT (subclavian vein or above) | Absolute indication for therapeutic anticoagulation |
| Active cancer | High risk of recurrent VTE if left untreated[36] |
| History of ≥1 prior episode of unprovoked VTE (±thrombophilia)[37] | High risk of recurrent VTE after stopping anticoagulation[38–40] |
| Clinical instability (systolic blood pressure <100 mm Hg or arterial oxygen saturation <92% at ambient air) at the time of presentation | Risk of clinical deterioration[4 14 41] |
| Active bleeding or at high risk of bleeding (eg, signs of active bleeding, ischaemic stroke during preceding <10 days,[42] major gastrointestinal bleeding during preceding <3 months, intracranial or intraocular bleeding <6 months,[42] major trauma or surgery during preceding <1 month,[42 43] platelets <75 ×10^9/L [3 44] or double anti-platelet therapy at the time of enrolment) | Contraindication to rivaroxaban |
| Severe renal failure (creatinine clearance <30 mL/min) | Contraindication to rivaroxaban |
| Severe liver insufficiency (Child-Pugh B and C) | Contraindication to rivaroxaban |
| Concomitant use of strong CYP3A4 inhibitors (ie, HIV protease inhibitors (saquinavir, indinavir, ritonavir, nelfinavir, amprenavir, lopinavir, atazanavir, fosamprenavir, tipranavir, darunavir), systemic azole antifungals (ie, ketoconazole, itraconazole, voriconazole or posaconazole)) or strong CYP3A4 inducers (ie, rifampicin, rifabutin, rifapentin, phenytoin, phenobarbital, primidone, carbamazepine or St. John's Wort) | Contraindication to rivaroxaban |
| Known hypersensitivity to rivaroxaban | Contraindication to rivaroxaban |
| Need for therapeutic anticoagulation for another reason (atrial fibrillation/flutter, mechanical heart valves, previous VTE, known antiphospholipid antibody syndrome with unprovoked VTE) | Randomisation to placebo unethical |
| Therapeutic anticoagulation for >72 hours for any reason at the time of screening | Could confound study outcomes |
| Hospitalised for >72 hours prior to the diagnosis of isolated SSPE (hospital-acquired VTE) | Could confound study outcomes due to influence of cotreatments |
| Known pregnancy or breast feeding | Contraindication to rivaroxaban |
| Lack of safe contraception in women of childbearing potential | Rivaroxaban is contraindicated in pregnancy |
| Refusal or inability to provide informed consent | Unethical |
| Prior enrolment in this trial | Confounds study outcomes |

DVT, deep vein thrombosis; VTE, venous thromboembolism

has convincingly shown that in patients with suspected DVT in whom DVT has been excluded by a single whole-leg CUS and in whom anticoagulation is withheld, the risk of recurrent VTE is very low.[49]

### Intervention

Patients in the intervention group receive clinical surveillance and a matching rivaroxaban placebo orally using the same dosing schedule, frequency of administration and duration of treatment as for rivaroxaban (see later). Clinical surveillance is done during the follow-up interviews by phone or in-person, depending on local practice. At each contact, trained study personnel complete an assessment of symptoms and review for suspected recurrent VTE and bleeding. If patients report symptoms or signs suggestive of recurrent VTE or significant bleeding, they are asked to present immediately to an emergency department (ED) for evaluation.

### Control

Patients who are assigned to the anticoagulation group receive oral rivaroxaban 15 mg two times per day for the first 21 days, followed by 20 mg once daily for the remaining 90-day study period. After completion of the treatment period, the study drug is discontinued. The same surveillance schedule as in the intervention group is used.

### Study procedures

Patients are screened for eligibility if they receive a CTPA in the ED or within 72 hours of hospitalisation to rule-in or rule-out PE, or if they present to an outpatient service within 72 hours of SSPE diagnosis. Eligibility criteria are assessed for all consecutive patients with suspected SSPE, and potentially eligible patients are asked to provide informed consent by investigators or their delegates (online supplemental file 1 for an English language

example of the informed consent form). For those who consent to participate, a bilateral whole-leg CUS is performed by qualified examiners (eg, radiologists, vascular specialists or emergency physicians). All eligible and consenting patients without DVT are randomised to receive clinical surveillance or anticoagulation with rivaroxaban, and trained study personnel collect baseline data by reviewing medical records, medication lists, laboratory results that have been obtained as part of routine care, radiology reports and by patient interview (table 3). After provision of the assigned study medication, the participants are asked to immediately start with the treatment. In addition, all participants receive a patient diary consisting of two parts: the first contains important information about the study, and the second part consists of a diary to record dates and type of outcomes in order to minimise recall bias (table 3).

Participants are followed for 90 days using phone or in-person interviews at 10, 30 and 90 days after randomisation (online supplemental file 1). Trained study personnel contacts patients, family members, and/or primary care physicians and review medical charts to obtain information about outcome events. At the end of the treatment period, patients are instructed to return the medication bottles and the patient diary. Drug-adherence is assessed by counting the pills in the returned medication bottles (table 3).

## Data collection and quality

Data are collected using an electronic database (secuTrial). The following measures are implemented to ensure optimal data quality and completeness: (1) training of study personnel in the methods of data abstraction, patient inquiry and data recording, (2) recording of study data on standardised electronic case report forms, (3) operations manual providing information on definitions and acceptable data sources for all variables, (4) central data monitoring with generation of statistical reports and individual data checks and (5) risk based on-site monitoring (ie, based on key performance indicators such as inappropriate recruitment rate, change of principal investigator, high number of queries raised by central data monitoring, high number of protocol deviations, etc).

## Criteria for discontinuation of the study medication and unblinding

In participants requiring prophylactic or therapeutic anticoagulation during the study period for reasons other than the index SSPE or if treatment with another prohibited agent is necessary (ie, strong CYP3A4 inhibitors or inducers, dual antiplatelet therapy, GP IIb/IIIa inhibitors), the study drug should be temporarily interrupted and restarted as soon as possible following discontinuation of the prohibited medication. Patients who develop any condition requiring permanent therapeutic anticoagulation (eg, atrial fibrillation) should permanently discontinue the study drug. If an invasive procedure or surgical intervention is required, the study medication should be stopped at least 24 hours prior to an elective intervention, or immediately for emergency procedures. If objectively confirmed recurrent VTE or pregnancy is diagnosed during the treatment period, the study drug should be discontinued and treatment allocation unblinded. Similarly, in case of major bleeding, the study drug should be stopped and unblinding may be necessary if emergency anticoagulation reversal is indicated.

## Primary and secondary outcomes

The primary (efficacy) outcome is the proportion of recurrent, clinically symptomatic, objectively confirmed VTE within 90 days of randomisation, defined as recurrent PE or lower limb DVT.[50 51] The objective diagnostic criterion for PE, based on available radiographic reports, is a new intraluminal filling defect on CTPA or pulmonary angiography; a perfusion defect involving at least 75% of a segment, with corresponding normal ventilation (ie, high probability lung scan); the confirmation of new PE at autopsy; or objectively confirmed proximal DVT of the lower extremity in patients with symptoms of PE. The objective diagnosis of DVT is the non-compressibility of a venous segment on CUS or an intraluminal filling defect on contrast venography. Because compression of iliac veins and the inferior vena cava may be technically difficult, additional diagnostic criteria for iliac and caval DVT also include abnormal duplex flow patterns compatible with thrombosis or an intraluminal filling defect on CT or MRI venography.[52] Both proximal and distal DVTs are considered.

Separate secondary (safety) outcomes include the proportion of clinically significant bleeding and all-cause mortality 90 days following randomisation. Clinically significant bleeding is a composite endpoint of major and clinically relevant non-major bleeding. Major bleeding is defined as fatal bleeding, symptomatic bleeding at critical sites (intracranial, intraspinal, intraocular, retroperitoneal, intra-articular, pericardial or intramuscular with compartment syndrome), or bleeding with a reduction of haemoglobin $\geq 20$ g/L, or bleeding leading to transfusion of $\geq 2$ units of packed red blood cells according to the definition of the International Society on Thrombosis and Haemostasis.[53] Clinically relevant non-major bleeding is defined as overt bleeding that does not meet criteria for major bleeding but is associated with a medical intervention, unscheduled physician contact (visit or telephone call), temporary cessation of the study drug, pain or impairment of activities of daily life.[54]

Information about the date, type and circumstances of outcome events is obtained from patients, family members, and/or healthcare providers during the follow-up interviews or by reviewing medical charts. In patients who experience recurrent VTE, the radiographic report and images confirming VTE recurrence are obtained. For patients who died during follow-up, the cause of death based on medical reports, death certificates and autopsy reports (if available) is recorded. All medical outcome

**Table 3** Study schedule

| Study period | Enrolment and allocation | Baseline | Post-allocation | | Close-out |
|---|---|---|---|---|---|
| Visit/follow-up phone call | 1 | | 2☏††‡‡ | 3☏†† | 4☏†† |
| Time (day with allowed visit window) | d0 | d0 | d10 (7–12) | d30 (28–35) | d90 (88–95) |
| Eligibility screen (inclusion/exclusion criteria) | x | | | | |
| Patient information and informed consent | x | | | | |
| Bilateral whole-leg compression ultrasonography* | x | | | | |
| Randomisation | x | | | | |
| Demographic characteristics | | x | | | |
| Risk factors for venous thromboembolism | | x | | | |
| Symptoms of venous thromboembolism | | x | | | |
| Comorbid conditions | | x | | | |
| Physical examination findings | | x | | | |
| Laboratory test results | | x | | | |
| Imaging findings | | x | | | |
| Concomitant treatments | | x | | | |
| Health-related quality of life (PEmb-QoL)† | | x | | x | x |
| Functional status‡ | | | | | x |
| Treatment setting | | x | | | |
| Distribution and instruction of study drug | | x | | | |
| Daily intake of study medication: placebo or rivaroxaban | | x (immediate start on day 0) | | | |
| Instructions and distribution of patient diary§ | | x | | | |
| Recurrent venous thromboembolism | | | x | x | x |
| Clinically significant bleeding | | | x | x | x |
| All-cause mortality | | | x | x | x |
| Medical resource utilisation | | | x | x | x |
| Time to symptom resolution | | | x | x | x |
| New concomitant treatments | | | x | x | x |
| Interruption of the study drug | | | x | x | x |
| Adherence¶ | | | | | x |
| Serious adverse events reporting** | | x (immediate start after inclusion on day 0) | | | |

*Participants with concomitant deep vein thrombosis are excluded from further study participation (screening failures).

†At the baseline visit, participants are asked to fill in the PEmb-QoL questionnaire. For follow-up assessments, participants receive the PEmb-QoL including a prestamped envelope by mail and are asked to complete and return the questionnaire. If the follow-up interview is done in-person, the PEmb-QoL can be administered during the office visit.

‡ Functional status is assessed using the post-venous thromboembolism functional status scale.

§The first part of the patient diary consists of an information on the study outline, surveillance interviews, and symptoms and signs suggestive for recurrent venous thromboembolism and bleeding, the investigator's contact information/emergency telephone number if these symptoms/signs occur, how to take the study medications, and instructions to return the drug bottles at the end of the study. The second part consists of a patient diary where patients are asked to record dates and type of outcomes and measures of health resource utilisation (eg, physician visits), and the time to symptom resolution and return to work/usual activities.

¶Adherence is assessed by counting the pill count of the returned medication bottles.

**A serious adverse event is defined as any untoward medical occurrence that results in in death, is life-threatening, requires inpatient hospitalisation or prolongation of existing hospitalisation, results in persistent or significant disability/incapacity, or is a congenital anomaly/birth defect. In addition, important medical events that may jeopardise the patient or may require an intervention to prevent one of these outcomes is also considered serious.

††Phone or in-person follow-up, depending on local practice.

‡‡See online supplemental file 2 for the case report form of the 10-day follow-up interview as an illustration of the content of the follow-up phone calls.

PEmb-QoL, Pulmonary Embolism Quality of Life.

events are reviewed and adjudicated by a committee of three independent clinical experts unaware of treatment assignment. Based on available information, death is adjudicated as PE-related, due to major bleeding (any death following an intracranial haemorrhage or a bleeding episode leading to haemodynamic deterioration), due to another cause or due to an undetermined cause. Death is considered PE-related in the following situations: (1)

autopsy-confirmed PE in the absence of another more likely cause of death, (2) objectively confirmed PE within the last 48 hours before death in the absence of another more likely cause of death or (3) PE is not objectively confirmed, but is most likely the main cause of death.[55] Final classification of all medical outcomes is based on the full consensus of the committee.

## Ancillary outcomes

Ancillary outcomes include health-related quality of life and functional status, both important patient-centred outcomes and medical resource utilisation, an economically relevant efficiency measure. Disease-specific, health-related quality of life at baseline, 30 and 90 days after the index PE is assessed using the Pulmonary Embolism Quality of Life (PEmb-QoL) questionnaire.[56 57] The PEmb-QoL is a validated, self-administered 40-item questionnaire to quantify quality of life in patients having experienced PE. Functional status is measured at 90 days after randomisation using the post-VTE functional status scale.[58 59] This scale has been proposed to assess functional limitations after VTE, covering aspects of daily life (including limitations in usual activity and changes in lifestyle) that are affected by the consequences of VTE and its complications. The assessment is done using a short, structured interview 90 days after randomisation.

The following measures of medical resource utilisation and productivity are assessed[44 60]: initial length of stay (LOS), subsequent overall hospitalisations as well as overall ED and physician outpatient visits within 90 days of randomisation, and time to return to work in workers or usual activities (eg, household) in non-workers. Information on these outcomes are obtained from the participant during the follow-up interviews, by interview of the patient's primary care physician and by hospital chart review. Subsequent healthcare contacts are classified as potentially related to VTE if a patient had chest or leg symptoms or signs (dyspnoea, chest pain, pleural effusion or leg pain or swelling), or bleeding complications.

## Withdrawals and loss to follow-up

Given the low treatment and follow-up burden and the short follow-up period of 90 days, we expect that completeness of follow-up data collection will be close to 100% based on prior experience.[44] A patient who withdraws consent for follow-up at 10 or 30 days may still agree with passive follow-up (ie, that study personnel may collect follow-up information from medical records of the participant's primary care physician) or to continue with the assessment at 90 days, if given the option. If study withdrawal occurs, data collected up to the time of withdrawal is used in a coded manner. If a participant is lost to follow-up, primary care physicians and surrogates are contacted, and the hospital records are consulted to obtain information about primary and secondary outcomes and survival status.

## Post-hoc evaluation of radiological interobserver agreement

CTPA images undergo central review at the Bern University Hospital by a panel of two experienced thoracic radiologists blinded to the interpretation of the radiologist at the enrolling site. Based on the consensus of this panel, the initial CTPA readings are classified into three categories: (1) presence of isolated SSPE (true-positive isolated SSPE), (2) absence of any PE (false-positive isolated SSPE) and (3) presence of segmental, lobar or central PE (false-negative higher level PE). The panel also evaluates the technical quality of the CTPA examination (adequacy of opacification, breathing artefacts) and the number of filling defects (ie, single vs multiple isolated SSPE). Confirmation of the SSPE diagnosis prior to enrolling the patient is logistically not feasible due to the short timeline between diagnosis and enrolment, and it would not reflect real-world practice, thus limiting the external validity of our study results. Therefore, the central review of CTPA images is done post-hoc in 6-month batches.

## Sample size calculation

Assumptions on VTE recurrence risk are based on data from 127 low-risk patients with isolated SSPE who received anticoagulants (warfarin or low-molecular-weight heparin), showing a VTE recurrence risk of 0.8% at 90 days after diagnosis.[23] We chose an absolute non-inferiority margin of 3.5% on the basis of recruitment feasibility, clinical acceptability and previous studies. This corresponds to a difference which is considered acceptable by most physicians and patients for the following reasons. First, our margin is within the range of the 3-month VTE recurrence proportion (0.5%–5%) below which thrombosis specialists would not initiate anticoagulation for PE.[61] Second, the definition of a clinically acceptable non-inferiority margin for recurrent VTE must also take into account the potential benefits of withholding anticoagulation, that is, the substantially lower risk of clinically significant bleeding (<1% vs 7% within 3 months for patients receiving anticoagulants).[5 23 25 62] Indeed, a patient group with PE who was involved in the trial planning process indicated that given the bleeding risk associated with anticoagulants, a VTE recurrence proportion of <5% seemed acceptable. Finally, similar non-inferiority margins (3%–5%) have been used in key studies comparing different drug treatment regimens and inpatient versus outpatient management for acute VTE.[42 44 63–67]

To determine the sample size, we used a Monte-Carlo simulation approach based on an Agresti-Caffo CI for risk difference.[68] Assuming a baseline VTE recurrence proportion of 1.0% at 90 days in both treatment groups, an absolute margin of 3.5% defining non-inferiority for clinical surveillance and a sampling ratio of 1:1 allowing 5% attrition (dropouts, including patients who died from non-VTE-related causes) in each group during 90 days, we estimated that 276 patients (138 per group) would result in at least 80% power to establish non-inferiority at an one-sided type I error of 5%.

## Planned statistical analyses

As recommended by the Consolidated Standards of Reporting Trials statement for non-inferiority trials,[69] we will perform both intention-to-treat (ITT) and per-protocol (PP) analyses. For the primary outcome, ITT and PP analyses should reach the same conclusion to consider results to be robust. For the remaining outcomes, the ITT analysis will be the primary analysis, the PP analysis a secondary analysis.

In the ITT analysis, all randomised patients will be analysed within the treatment group to which they were randomised. In the PP analysis, patients with protocol violations will be excluded (crossover to study treatment from other group; patients receiving a different type or dose of anticoagulation than requested by the study protocol; patients with missing primary outcome data; patients violating relevant eligibility criteria; patients stopping study treatment within 1 month after randomisation, or patients in whom the diagnosis of isolated SSPE was refuted by the central CTPA review panel).

We will describe the prevalence of recurrent VTE, clinically significant bleeding (including its individual components, major and clinically relevant non-major bleeding), and all-cause mortality at 90 days after randomisation with 95% Wilson CIs by treatment group. Non-inferiority of the primary outcome (VTE recurrence) among patients in the clinical surveillance versus the anticoagulation group will be assessed based on the Agresti-Caffo CI for risk difference.[68] If the upper limit of the one-sided 95% Agresti-Caffo CI will be lower than the prespecified non-inferiority margin, clinical surveillance will be considered non-inferior to anticoagulation. For secondary outcomes, we will calculate the risk difference with a two-sided 95% Agresti-Caffo CI and compare groups using an exact binomial test.[68]

PEmb-QoL dimension scores at 30 and 90 days as well as the change in scores from baseline will be presented by treatment group as means with 95% CIs. The change in dimension scores and the differences in the change between groups will be analysed using a repeated-measures, linear mixed-effects model adjusted for the respective baseline value. Functional status, which is measured on a scale from 0 (no functional limitations) to 5 (death) at 90 days, will be presented as median and IQR and compared between groups using the non-parametric Wilcoxon rank-sum test. Count data (subsequent hospitalisations, outpatient visits) will be presented by treatment group as rate with an exact 95% Poisson CI and compared using a rate ratio and exact p value. Time-to-event outcomes (initial LOS, time to return to work/usual activities) will be presented as medians and IQRs. For LOS, we will compare groups using the Wilcoxon rank-sum test. For return to work, we will display Kaplan-Meier curves and compare groups by the log-rank test. An alpha level of <0.05 will define statistical significance.

In secondary analyses, we will calculate the cumulative incidence and the difference in cumulative incidence of VTE recurrence and clinically significant bleeding, correcting for withdrawals and losses to follow-up by censoring and for death unrelated to VTE recurrence or bleeding as a competing event. All-cause mortality will be assessed likewise, however, a competing event does not apply. In a further secondary analysis, we will use model-based approaches for all outcomes. For time-to-event outcomes (VTE recurrence, clinically significant bleeding, LOS, return to work/usual activity), we will use competing risk regression according to Fine and Gray,[70] accounting for non-VTE/non-bleeding-related death or all-cause death as a competing event. For mortality, we will use Cox regression. For count data, we will use a negative binomial model, and also consider zero-inflation. In case of heterogeneity across sites, we will adjust models for site using random-effects models.

## Data monitoring and interim safety analysis

An independent Data and Safety Monitoring Board (DSMB) consisting of three members unaffiliated with any of the participating institutions will evaluate unblinded interim safety results after 100 and 190 randomised patients have completed the 90-day follow-up. To monitor recurrent VTE in the intervention group, formal interim analyses will be performed using a Bayesian approach. Based on literature,[23] we expect a frequency of VTE recurrence of 1.0% (0%–4.5%). We regard a frequency of >5% in the surveillance group as clinically unacceptable. We will calculate the prior probability distribution of events based on the expected frequency as well as the posterior probability distribution using actually observed data at each interim analysis. Based on the posterior distribution, we will calculate the probability that the true proportion of VTE recurrence in the surveillance group exceeds the threshold of unacceptable frequency. If the probability of exceedance is >70%, the surveillance arm will be considered inferior and the DSMB will recommend to stop the trial. The final decision will lie with the Steering Committee. This procedure is conservative regarding the type I error rate; therefore, we will not adjust the significance level in the final analysis.

## Patient and public involvement

We have partnered with a group of patients who had recently experienced PE in the planning process of this trial, including the selection of patient-centred outcomes, the establishment of a safe surveillance schedule and the determination of a clinically acceptable non-inferiority margin. Patients are not involved in the recruitment and conduct of the study or the dissemination of study results.

## Data sharing

After publication of the study results, a de-identified patient-level data set relating to the primary publication along with the latest version of the study protocol, the informed consent form, the statistical analysis plan, the analysis code and the data management plan of the study will be made publicly available in the *Bern Open Repository and Information System* (*BORIS*) *Research Data*.

## ETHICS AND DISSEMINATION

This trial is conducted in accordance with the Declaration of Helsinki, the International Council on Harmonization Good Clinical Practice guidelines, and all applicable national legal and regulatory requirements. Authorisation by the local Ethics Committees and Swissmedic has been obtained in Switzerland, and the submission process to the relevant Dutch and Canadian Ethics Committees and regulatory authorities is ongoing. All changes in research activity or unanticipated problems involving risks to human subjects will be reported promptly to the ethics committees. All participating investigators/institutions will permit study-related monitoring, audits, ethics committee reviews and regulatory inspections, and will provide direct access to source documents and data. Protection of confidentiality is ensured according to regulatory requirements.

Study personnel informs eligible patients with a diagnosis of isolated SSPE about all aspects of the study participation, including the goals, procedures and potential risks/benefits associated with the study. Potential participants are informed that the decision to participate in the study is entirely voluntary and that they may withdraw from the study at any time, with no effect on their current or future treatment. Written informed consent is obtained from all eligible patients prior to any study-related procedure, including bilateral whole-leg CUS. The primary study results will be presented at scientific conferences and published in a peer-reviewed medical journal. We also plan local presentations for physicians at the participating sites. Participants will receive a letter with the study results explained in lay language. Further public dissemination of the results is planned through publications in the lay press and via social media.

## CONCLUSION

The SAFE-SSPE trial addresses an important gap of knowledge, the optimal management of SSPE, and represents the first randomised, direct comparison of clinical surveillance alone versus anticoagulant treatment for low-risk patients with isolated SSPE. As the number of SSPE patients who may eventually receive anticoagulant treatment is likely to rise in the future with the further dissemination and advancement of CTPA technology, a strong scientific basis for withholding anticoagulation in low risk patients with SSPE is urgently needed. The results of this trial have the potential to improve quality and efficiency of care by reducing bleeding episodes and resource utilisation and increasing health-related quality of life.

### Current status of the SAFE-SSPE trial

Patient recruitment has started in May 2020; by July 2020, six patients have been enrolled in the trial. Follow-up of the last participant is expected to be completed in February 2024.

### Author affiliations
[1]Department of General Internal Medicine, Inselspital, Bern University Hospital, University of Bern, Bern, Switzerland
[2]Department of Thrombosis and Hemostasis, Leiden University Medical Center, Leiden, The Netherlands
[3]Department of Medicine, The Ottawa Hospital Research Institute at the University of Ottawa, Ottawa, Ontario, Canada
[4]CTU Bern, University of Bern, Bern, Switzerland
[5]Division of Angiology and Haemostasis, Geneva University Hospital, University of Geneva, Geneva, Switzerland
[6]Department of Internal Medicine, Cantonal Hospital of Baden, Baden, Switzerland

**Acknowledgements** We would like to thank Sven Trelle, Director of the CTU Bern (Clinical Trials Unit) for his methodological inputs and organisational work concerning this study.

**Contributors** Study concept and design: DA, MVH, MC, FAK, J-HB, MR, AL, CB. Drafting of the manuscript: CB, DA. Critical revision of the manuscript for important intellectual content: MVH, MC, FAK, JHB, MR, AL, JM, MP, DR. Statistical analysis plan: AL. Obtained funding: DA, MR, J-HB. Administrative, technical or material support: MP, JM, DR, CB. Supervision: DA.

**Funding** This study is supported by an Investigator Initiated Clinical Trials (IICT) grant of the Swiss National Science Foundation (SNSF 33IC30_185616). The study medication was provided free of charge by Bayer AG.

**Competing interests** MC reports grants from Leo Pharma, BMS, Pfizer, personal fees from Bayer, BMS, Pfizer, Sanofi, Leo Pharma, Servier, outside the submitted work. All other authors declare that they have no competing interest with regard to the intellectual concern and proprietary of affairs. The funding source had no role in the design of this study and does not have any role in the conduct, the collection, management, analysis and interpretation of the data, the writing of the manuscript or the decision to submit the results for publication.

**Patient consent for publication** Not required.

**Provenance and peer review** Not commissioned; externally peer reviewed.

**ORCID iD**
Christine Baumgartner http://orcid.org/0000-0003-2296-9632

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
