## [Reviewer comments · BMJ Open]

ARTICLE DETAILS

TITLE (PROVISIONAL)	Clinical Surveillance vs. Anticoagulation For low-risk patiEnts with isolated SubSegmental Pulmonary Embolism: Protocol for a multicenter randomized placebo-controlled non-inferiority trial (SAFE-SSPE)
AUTHORS	Baumgartner, Christine; Klok, Frederikus; Carrier, Marc; Limacher, Andreas; Moor, Jeanne; Righini, Marc; Beer, Jürg-Hans; Peluso, Martina; Rakovic, Damiana; Huisman, Menno; AUJESKY, Drahomir

VERSION 1 – REVIEW

REVIEWER	Adam Singer Stony Brook University, USA
REVIEW RETURNED	03-Jul-2020

GENERAL COMMENTS	GENERAL COMMENTS The proposed protocol is well designed and well written and addresses a very important issue: whether anticoagulation is necessary for isolated subsegmental PE in low risk patients. I look forward to the results of this trial. SPECIFIC COMMENTS Article summary: Would add to the third bullet “as well as costs”. Introduction. Page 6, Line 43. Citations should be 19-21. Selection of patients. It is well known that interobserver reliability of diagnosing SSPE is fair. Should inclusion criteria require that all CTs be reviewed by at least 2 radiologists before enrolling the patient? Many experts recommend confirming diagnosis before considering treatment. Intervention. It might be helpful to submit an appendix with the structured telephone survey at follow up. Data collection. What do you mean by “risk based on-site monitoring”? Outcomes. It is not clear if distal DVT is also considered as an outcome. In one place (P12, L 29) it sates proximal DVT, while in another (P12, L40) it includes both distal and proximal DVTs. Planned analyses. P17, L21. Please clarify the functional status. Is 0 to 5 from best to worst or worsts to best?
---

	Current Status. Change “Mai” to “May”. Since it is now July, has any patient been enrolled to date?
--	---

REVIEWER	Hugo Hyung Bok Yoo Botucatu Medical School of São Paulo State University-UNESP, Brazil
REVIEW RETURNED	07-Jul-2020

GENERAL COMMENTS	This study will be the first randomized and well-designed trial comparing the safety and efficacy of a management strategy without anticoagulation and anticoagulant treatment in low-risk patients with isolated subsegmental pulmonary embolism (SSPE). However, this Reviewer has some minor comments and questions:  • “Low-risk patient” needs to be better defined; • Are thrombophilic patients included or excluded from the study? • Ethics: it recommended describe how the investigators plan to get the informed consent process from the participants; • When using drugs, both scientific and brand name should be mentioned followed by the name of the manufacturing company, city, and country. Drug route, dosage, frequency of administration, and total duration of treatment with the drug should be clearly mentioned; • Involved personnel should precisely define: who will be responsible for the interventions? What activities each personnel will perform and with what frequency and intensity during the study?
---

REVIEWER	Waldemar E. Wysokinski Mayo Clinic Rochester, MN, USA
REVIEW RETURNED	20-Jul-2020

GENERAL COMMENTS	Pulmonary embolism (PE) is an acute complication of deep vein thrombosis and is the third most common cause of cardiovascular death (1-3). PE, however, encompasses a wide range of presentations and prognoses ranging from asymptomatic to hemodynamic instability and sudden death. Based on the clinical presentation, PE can be divided into symptomatic or asymptomatic and based on anatomic criteria, bilateral or unilateral and by a caliber of the pulmonary artery obstructed as saddle, lobar, segmental, or subsegmental. Several prognostic stratification tools containing a mixture of clinical, laboratory, and anatomical findings were proposed based on their impact on PE clinical outcomes, particularly mortality (4,5). As PE categories are reflecting predicted outcomes they consequently determine specific therapeutic strategies that range from no anticoagulation, anticoagulation alone, catheter-directed thrombolysis, full-, or reduced dose systemic thrombolysis, catheter-based or surgical embolectomy, and mechanical circulatory support such as extracorporeal membrane oxygenation. PE with embolic material limited to sub-segmental pulmonary arteries called isolated subsegmental PE (ISSPE) is considered to be of low risk. (6-7) and if not associated with DVT and particularly, if in an asymptomatic patient with good cardiorespiratory reserve is considered as not requiring treatment with anticoagulant (10). However, the decision to offer anticoagulation therapy to a patient with ISSPE is both patient-specific and provider-dependent, requiring the clinician to weigh the perceived risk of disease-
--

	related morbidity against the risk of drug-related complications such as bleeding. Some studies have shown that it might be reasonable to manage patients with ISSPE without anticoagulation (11-13), others have not (7). Moreover, advance in PE diagnosis, such as the use of d-dimer measurement and particularly the introduction of multi-detector row computer tomography pulmonary angiography that allow detection of very small embolism raises now concern that the over-diagnosis of clinically less significant PE may be contributing to higher rates of treatment-related complications (3,7,14,15). Clinical and laboratory-based criteria on one hand and on the other hand criteria strictly based on radiographic findings, such as one used for ISSPE, can result in considerable overlap in the diagnosis of PE categories. Patients with ISSPE based on embolus location may represent PE clinically category of sub-massive events (16). Baumgartner et al. now report in the Journal the rationale and design of the international, multicenter, randomized, placebo-controlled, double-blind, parallel-group non-inferiority trial SAFE-SSPE (Surveillance vs. Anticoagulation For low-risk patiEnts with isolated SubSegmental Pulmonary Embolism) which is conducted in Switzerland, the Netherlands, and Canada. Low-risk patients with isolated SSPE are randomized to receive clinical surveillance with either placebo (no anticoagulation) or rivaroxaban. The primary outcome is symptomatic recurrent VTE (efficacy) and the secondary outcomes include clinically significant bleeding (composite of major bleeding and clinically relevant non-major bleeding) and all-cause mortality (safety). The ancillary outcomes are health-related quality of life, functional status, and medical resource utilization. This study is very relevant and clinically necessary as it addresses an important gap of knowledge, the optimal managing of SSPE, comparing no therapy versus anticoagulant treatment with widely used anticoagulant, rivaroxaban in a randomized fashion, for low-risk patients with isolated SSPE. There are, however, important aspects of this study design that may jeopardize the performance of this trial and might not allow obtaining meaningful results. First, there are unclear criteria for patient recruitment to the study. ISSPE is defined purely based on the anatomical distribution of radiological findings irrespectively of symptomatology and this study recruits asymptomatic as well as symptomatic patients. Since most patients with the diagnosis of asymptomatic ISSPE had CTPA done for other medical problems most often cancer, one can expect that a substantial proportion of patients classified as “low risk” would have symptomatic ISSPE. This group will include those with severe chest pain and shortness of breath related to pleural irritation by peripheral embolic material. Moreover, from the perspective of anatomical involvement by embolic material, this PE category encompasses cases with single isolated subsegmental embolic material as well as multiple, bilateral subsegmental emboli. It will be rather difficult to justify the randomization of very symptomatic patients with multiple, bilateral ISSPE to the placebo group. These patients cannot be excluded based on the “presence of cardiopulmonary compromise” because they do not have a cardiopulmonary compromise. Even, if we assume that severe pleuritic chest pain represents the “cardiopulmonary compromise” – would mild, intermittent, only with a deep breath pain, disqualify the patient? So, what would be the threshold that would still allow study participation? It introduces an important bias to the study.
--	---

The other problem related to patient recruitment could be the radiological accuracy of ISSPE diagnosis. Given the low interobserver agreement ($\kappa = 0.38$) on CTPA interpretation, even by experienced chest radiologists (17) there could be substantial variability in the accuracy of ISSPE diagnosis at the participating centers. It is rather unfortunate that the central review of CTPA images will be done post-hoc in 6-month batches. Disagreement in the false positive or negative detection of ISSPE could be a problem and may substantially affect the performance of the study particularly that only 276 patients (138 per group) are planned to be recruited. It would be much better if the central review at the Bern University Hospital would be performed within the first few days of patient recruitment.

The calculation of risk and benefit for this study is very narrow. The VTE recurrent rate is estimated by the authors to be 0.8%. The rate of major bleeding, as assessed by EINSTEIN-PE study (18), will be very close to 0.8% too. In this study, the rate of major bleeding in the rivaroxaban arm was 1.1% but only 5.3% of patients were treated for 90 days, 57.3% for 6 and 37.4% for 12 months. The rate of a composite of major and CRNMB was 10.3, but based on K-M curve analysis about half of bleeding events happened with the first 90 days. Moreover, one should expect bleeding events to be smaller in the group of "low risk" patients eligible for the current study.

Adding "all-cause mortality" to clinically significant bleeding as the safety outcomes introduce the factor which has little to do with study intervention. Low-risk patients have very little risk of death within 90 days and it is unlikely that this event would have anything to do with the rivaroxaban effect but will be rather governed by chance.

DVT of high potential for PE will not be detected by the current study protocol. This will include DVT of iliac and inferior vena cava thrombosis (25% of DVT) (19,20) of thrombosis of atypical location such as hepatic or gonadal veins, and cerebral venous sinus thrombosis (4% of DVT) (21). Although thromboses of these locations were not included in large randomized clinical trials of acute VTE this study has placebo arm and may put patients with undetected DVT at the substantial risk of lethal PE.

Based on the study description it appears that investigators will not accept magnetic resonance imaging venography to diagnose new PE events for VTE recurrence detection (only CTPA and ventilation-perfusion scan), but would accept magnetic resonance imaging venography technique to diagnose DVT. This seems to be strange and requires clarification.

REFERENCES

1. Tapson VF. Acute pulmonary embolism. *N Engl J Med* 2008;358(10):1037–1052.
2. Sista AK, Kuo WT, Schiebler M, Madoff DC. Stratification, Imaging, and Management of Acute Massive and Submassive Pulmonary Embolism. *Radiology*. 2017;284(1):5-24.
3. Heit JA, Ashrani AA, Crusan DJ, McBane RD, Petterson TM, Bailey KR. Reasons for the persistent incidence of venous thromboembolism. *Thrombosis and haemostasis*. 2017;117(02):390-400.
4. Jaff MR, McMurtry MS, Archer SL, et al. Management of massive and submassive pulmonary embolism, iliofemoral deep vein thrombosis, and chronic thromboembolic pulmonary

hypertension: a scientific statement from the American Heart Association. *Circulation*. 2011;123(16):1788-1830.

5. Konstantinides SV, Meyer G, Becattini C, et al. 2019 ESC Guidelines for the diagnosis and management of acute pulmonary embolism developed in collaboration with the European Respiratory Society (ERS). *Eur Heart J*. 2020 Jan 21;41(4):543-603. doi: 10.1093/eurheartj/ehz405.
6. Le Gal G, Righini M, Parent F, Van Strijen M, Couturaud F. Diagnosis and management of subsegmental pulmonary embolism. *J Thromb Haemost*. 2006; 4(4):724-31.
7. den Exter PL, van Es J, Klok FA, et al. Risk profile and clinical outcome of symptomatic subsegmental acute pulmonary embolism. *Blood*. 2013; 122(7):1144-1149.
8. Donze J, Le Gal G, Fine MJ, Roy PM, et al. Prospective validation of the Pulmonary Embolism Severity Index. A clinical prognostic model for pulmonary embolism. *Thromb Haemost* 2008;100(5):943–948.
9. Jimenez D, Aujesky D, Moores L, Gomez V, et al. Simplification of the pulmonary embolism severity index for prognostication in patients with acute symptomatic pulmonary embolism. *Arch Intern Med* 2010;170(15):1383–1389.
10. Kearon C, Akl EA, Ornelas J, et al. Antithrombotic Therapy for VTE Disease: CHEST Guideline and Expert Panel Report. *Chest*. 2016 ; 149(2):315-352.
11. Donato AA, Khoche S, Santora J, Wagner B. Clinical outcomes in patients with isolated subsegmental pulmonary emboli diagnosed by multidetector CT pulmonary angiography. *Thromb Res*. 2010; 126(4):e266-e270.
12. Stein PD, Goodman LR, Hull RD, Dalen JE, Matta F. Diagnosis and management of isolated subsegmental pulmonary embolism: review and assessment of the options. *Clin Appl Thromb Hemost*. 2012; 18(1):20-26.
13. Goy J, Lee J, Levine O, Chaudhry S, Crowther M. Subsegmental pulmonary embolism in three academic teaching hospitals: a review of management and outcomes. *J Thromb Haemost*. 2015;13(2):214-218.
14. Remy-Jardin M, Pistolesi M, Goodman LR, et al. Management of suspected acute pulmonary embolism in the era of CT angiography: a statement from the Fleischner Society. *Radiology*. 2007; 245(2):315-329.
15. Wiener RS, Schwartz LM, Woloshin S. Time trends in pulmonary embolism in the United States: evidence of overdiagnosis. *Arch Intern Med*. 2011; 171(9):831-837.
16. Cambron JC, Saba ES, McBane RD, et al. Adverse Events and Mortality in Anticoagulated Patients with Different Categories of Pulmonary Embolism. *Mayo Clin Proc Innov Qual Outcomes* 2020; 4(3):249-258.
17. Long B, Koyfman A. Best clinical practice: current controversies in pulmonary embolism imaging and treatment of subsegmental thromboembolic disease. *J Emerg Med*. 2017; 52(2):184-193.
18. Büller HR, Prins MH, Lensin AW, et al. EINSTEIN–PE Investigators. Oral rivaroxaban for the treatment of symptomatic pulmonary embolism. *N Engl J Med*. 2012; 366(14):1287–1297
19. Nyamekye I, Merker L. Management of proximal deep vein thrombosis. *Phlebology* 2012;27 Suppl 2:61-72.
20. Plate G, Ohlin P, Eklof B. Pulmonary embolism in acute iliofemoral venous thrombosis. *Br J Surg* 1985;72:912-5.

	21. Tafur AJ, Kalsi H, Wysokinski WE, et al. The association of active cancer with venous thromboembolism location: a population-based study. Mayo Clin Proc. 2011; 86(1):25-30.
--	--

VERSION 1 – AUTHOR RESPONSE

Comments from Reviewer 1:

SPECIFIC COMMENTS

1. Article summary: Would add to the third bullet “as well as costs”.

Done (see Article Summary on page 5).

2. Introduction. Page 6, Line 43. Citations should be 19-21.

Done (page 6, line 18).

3. Selection of patients. It is well known that interobserver reliability of diagnosing SSPE is fair. Should inclusion criteria require that all CTs be reviewed by at least 2 radiologists before enrolling the patient? Many experts recommend confirming diagnosis before considering treatment.

We agree that interobserver reliability of diagnosing SSPE is suboptimal. As this study aims to reflect real-world practice (where the physician needs to decide on the further management strategy once the SSPE diagnosis is confirmed by the local radiologist/s), the SSPE diagnoses are made according to usual practice at the participating sites. Moreover, although a central review of the CTPA images to confirm the SSPE diagnosis by a team of thoracic radiology experts would be ideal to ensure consistent SSPE diagnoses at all trial sites, this is logistically not feasible due to the short timeline between diagnosis and enrolment, and it would not reflect real-world practice, thus limiting the external validity of our study results. Therefore, the central review of CTPA images will be done post-hoc by 2 blinded thoracic radiologists. Patients in whom SSPE was not confirmed during central review will be excluded from the per-protocol analysis (see page 17, line 6). We have added this explanation to the revised Methods section on page 15, lines 18-21.

4. Intervention. It might be helpful to submit an appendix with the structured telephone survey at follow up.

We have now added a Supplementary File containing the case report form (CRF) for the first follow-up interview after 10 days to illustrate the content of the follow-up phone calls (mentioned on page 11, line 10, and in the footnote of Table 3, page 38).

5. Data collection. What do you mean by “risk based on-site monitoring”?

We have now specified in the Methods section on page 11, lines 22-24 that risk-based on-site monitoring refers to monitoring visits based on key performance indicators, such as inappropriate recruitment rate, change of principal investigator, high number of queries raised by central data monitoring, high number of protocol deviations, etc.

6. Outcomes. It is not clear if distal DVT is also considered as an outcome. In one place (P12, L 29) it states proximal DVT, while in another (P12, L40) it includes both distal and proximal DVTs.

The primary outcome is objectively confirmed VTE within 90 days of randomization, including PE, proximal and distal DVT. As mentioned on page 12, line 21 of the revised manuscript, objectively confirmed proximal DVT of the lower extremity in patients with symptoms of PE is one of the objective criteria for a PE outcome. However, both distal and proximal DVT will be considered for DVT outcomes.

7. Planned analyses. P17, L21. Please clarify the functional status. Is 0 to 5 from best to worst or worsts to best?

A score of 0 on the functional status scale refers to no functional limitations, while a score of 5 refers to death (i.e. worst functional status). We have clarified this in the revised Methods section on page 17, line 19.

8. Current Status. Change “Mai” to “May”. Since it is now July, has any patient been enrolled to date? We have now corrected the typographic error. We have updated the current status of the trial on page 21, lines 2-3, and have specified that we have enrolled 6 patients by July 2020.

Specific comments from Reviewer 2

This Reviewer has some minor comments and questions:

1. “Low-risk patient” needs to be better defined;

We now clarified that low risk patients refers to patients at low risk of recurrent or progressive VTE (revised Introduction section on page 6, line 23, and revised Methods section on page 8, lines 19-20). We have already specified that patients with a high early risk of VTE recurrence ($\geq 8\%$) are excluded from our study, which encompasses patients with active cancer, a history of prior unprovoked VTE, or those with hospital-acquired VTE (page 9, line 18; and Table 2).

2. Are thrombophilic patients included or excluded from the study?

Patients with thrombophilia are excluded from the study if they had a previous episode of unprovoked VTE, as this is an indication for extended anticoagulation treatment due to the high risk of VTE recurrence (see our reference 63). We have now specified this in Table 2 (page 36). Although patients with a first unprovoked VTE who have thrombophilia such as the antiphospholipid antibody syndrome (especially those based on the strict revised Sapporo criteria or who have different antiphospholipid antibody types) carry an increased risk of recurrence after stopping anticoagulation, the recurrence risk may be substantially lower in patients with provoked VTE (ref: Kearon C, et al. Blood 2018;131(19):2151-2160; Garcia D, et al. Blood. 2013;122(5):817-824). Thus, we will not exclude patients with thrombophilia including the antiphospholipid antibody syndrome per se but only those with prior unprovoked VTE.

3. Ethics: it recommended describe how the investigators plan to get the informed consent process from the participants;

We have now added a paragraph to the section on Ethics and Dissemination describing the informed consent process (page 20, lines 3-8):

“Study personnel informs eligible patients with a diagnosis of isolated SSPE about all aspects of the study participation, including the goals, procedures, and potential risks/benefits associated with the study. Potential participants are informed that the decision to participate in the study is entirely voluntary and that they may withdraw from the study at any time, with no effect on their current or future treatment. Written informed consent is obtained from all eligible patients prior to any study-related procedure, including bilateral whole-leg CUS.”

4. When using drugs, both scientific and brand name should be mentioned followed by the name of the manufacturing company, city, and country. Drug route, dosage, frequency of administration, and total duration of treatment with the drug should be clearly mentioned;

We now clearly mention drug route, dosage, frequency of administration, and total duration of treatment for both placebo and rivaroxaban in the revised Methods section (page 10, lines 6 and lines 14-15). The rivaroxaban used in this study is identical with Xarelto® except that it is a white tablet without any imprinting; thus, it is not as such a marketed product, but is exclusively produced for the use in placebo-controlled clinical trials. Therefore, we did not mention the brand name. The name of

the manufacturer (Bayer AG) is mentioned in the funding statement on page 3 and page 22; we did not list city and country of the manufacturer because several sites of the company are involved in the production and provision of the study drugs.

5. Involved personnel should precisely define: who will be responsible for the interventions? What activities each personnel will perform and with what frequency and intensity during the study?

We clarified in the revised Methods section on page 10, line 23-24 that informed consent is obtained by investigators or their delegates and that bilateral whole-leg compression ultrasonography (CUS) is performed by qualified examiners such as radiologists, vascular specialists, or emergency physicians (page 11, line 1). Trained study personnel collect baseline data (mentioned on page 11, line 3) and perform follow-up phone calls or in-person interviews 10, 30, and 90 days after randomization (page 11, line 10).

Specific comments from Reviewer 3

This study is very relevant and clinically necessary as it addresses an important gap of knowledge, the optimal managing of SSPE, comparing no therapy versus anticoagulant treatment with widely used anticoagulant, rivaroxaban in a randomized fashion, for low-risk patients with isolated SSPE. There are, however, important aspects of this study design that may jeopardize the performance of this trial and might not allow obtaining meaningful results.

1. First, there are unclear criteria for patient recruitment to the study. ISSPE is defined purely based on the anatomical distribution of radiological findings irrespectively of symptomatology and this study recruits asymptomatic as well as symptomatic patients. Since most patients with the diagnosis of asymptomatic ISSPE had CTPA done for other medical problems most often cancer, one can expect that a substantial proportion of patients classified as “low risk” would have symptomatic ISSPE. This group will include those with severe chest pain and shortness of breath related to pleural irritation by peripheral embolic material. Moreover, from the perspective of anatomical involvement by embolic material, this PE category encompasses cases with single isolated subsegmental embolic material as well as multiple, bilateral subsegmental emboli. It will be rather difficult to justify the randomization of very symptomatic patients with multiple, bilateral ISSPE to the placebo group. These patients cannot be excluded based on the “presence of cardiopulmonary compromise” because they do not have a cardiopulmonary compromise. Even, if we assume that severe pleuritic chest pain represents the “cardiopulmonary compromise” – would mild, intermittent, only with a deep breath pain, disqualify the patient? So, what would be the threshold that would still allow study participation? It introduces an important bias to the study.

Thank you for your comment. Our study has clearly defined eligibility criteria for patient recruitment (see inclusion criteria on page 9, lines 7-9, and exclusion criteria in Table 2).

Patients with both symptomatic and incidentally detected, asymptomatic SSPE are eligible for recruitment, because symptoms of PE may be subtle and difficult to elucidate, and because symptomatic and asymptomatic PE appear to have a similar prognosis (see our new reference 35). Specifically, we are unaware of an existing association between pleuritic pain severity (which is somewhat subjective) and prognosis. As we exclude patients with hospital-acquired PE or active cancer (in whom most incidental PEs are found), we expect to enroll only few patients with incidentally detected, asymptomatic SSPE (<2%; see our new reference 35 and Dentali F, et al. *Thromb Res.* 2010; 125(6):518-522). We enroll both patients with single and multiple isolated SSPE, because there is no convincing evidence that these conditions are prognostically different. We have now added this information to page 9, lines 13-17.

In addition, we have clarified that cardiopulmonary compromise relates to hypotension (systolic blood pressure <100mmHg) or hypoxemia (arterial oxygen saturation <92% at ambient air) at the time of presentation (page 9, line 19, and Table 2). Thus, pleuritic pain of any severity, if not associated with hypoxemia or a systolic blood pressure <100 mm Hg, is not a patient selection criterion.

2. The other problem related to patient recruitment could be the radiological accuracy of ISSPE diagnosis. Given the low interobserver agreement ($\kappa = 0.38$) on CTPA interpretation, even by experienced chest radiologists (17) there could be substantial variability in the accuracy of ISSPE diagnosis at the participating centers. It is rather unfortunate that the central review of CTPA images will be done post-hoc in 6-month batches. Disagreement in the false positive or negative detection of ISSPE could be a problem and may substantially affect the performance of the study particularly that only 276 patients (138 per group) are planned to be recruited. It would be much better if the central review at the Bern University Hospital would be performed within the first few days of patient recruitment.

Please see our answer to comment 3 from reviewer #1.

3. The calculation of risk and benefit for this study is very narrow. The VTE recurrent rate is estimated by the authors to be 0.8%. The rate of major bleeding, as assessed by EINSTEIN-PE study (18), will be very close to 0.8% too. In this study, the rate of major bleeding in the rivaroxaban arm was 1.1% but only 5.3% of patients were treated for 90 days, 57.3% for 6 and 37.4% for 12 months. The rate of a composite of major and CRNMB was 10.3, but based on K-M curve analysis about half of bleeding events happened with the first 90 days. Moreover, one should expect bleeding events to be smaller in the group of "low risk" patients eligible for the current study.

Given that we are studying a low-risk population, we agree with the Reviewer that the risk-benefit ratio of clinical surveillance vs. anticoagulation may be relatively small. In the EINSTEIN-PE study, from which patients with an increased bleeding risk were also excluded, the cumulative 90-day incidence of recurrent VTE was around 1.5%, whereas the cumulative 90-day incidence of clinically significant bleeding was about 7%. Still, if our study finds a low 90-day VTE recurrence rate in the clinical surveillance group without anticoagulation (we assumed a 0.8% recurrence based on a previous study of low-risk patients) and a lower bleeding rate than in the anticoagulation group (with is highly probable), it has the potential to improve quality of care and reduce treatment costs in patients with isolated SSPE.

4. Adding "all-cause mortality" to clinically significant bleeding as the safety outcomes introduce the factor which has little to do with study intervention. Low-risk patients have very little risk of death within 90 days and it is unlikely that this event would have anything to do with the rivaroxaban effect but will be rather governed by chance.

We apologize for not having indicated this clearly enough, but clinically significant bleeding and all-cause mortality are two separate secondary outcomes. We have clarified this on page 13, line 3.

5. DVT of high potential for PE will not be detected by the current study protocol. This will include DVT of iliac and inferior vena cava thrombosis (25% of DVT) (19,20) of thrombosis of atypical location such as hepatic or gonadal veins, and cerebral venous sinus thrombosis (4% of DVT) (21). Although thromboses of these locations were not included in large randomized clinical trials of acute VTE this study has placebo arm and may put patients with undetected DVT at the substantial risk of lethal PE. The reference cited by the reviewer refers to iliofemoral thrombosis (25% of DVT according to the reviewer's reference 19). However, isolated iliac or caval DVT is very uncommon, and the latter is mostly associated with the presence of inferior vena cava filters (ref: Jain AK, et al. J Vasc Surg Venous Lymphat Disord. 2018;6(6):724-729; Alkhouli M, et al. JACC Cardiovasc Interv 2016. 9(7):629-43). Similarly, splanchnic vein thromboses and cerebral vein thromboses are uncommon and not typically a source of embolization to the lungs, so it is unlikely that isolated SSPE arises from isolated thromboses at these sites (ref: Ageno W, et al. Blood 2014;124(25):3685-91; Liberman AL, et al. Stroke 2017;48(3):563-567). While guidelines recommend excluding proximal lower extremity DVT if a management strategy without anticoagulation is considered for SSPE, they do not specifically recommend to screen for thromboses at atypical locations (ref: CHEST Guideline and Expert Panel Report. Chest 2016; 149(2):315-352; 2019 ESC Guidelines, Eur Heart J 2020;41:543-603). Thus, our

method for detecting DVT is consistent with current guideline recommendations.

6. Based on the study description it appears that investigators will not accept magnetic resonance imaging venography to diagnose new PE events for VTE recurrence detection (only CTPA and ventilation-perfusion scan) but would accept magnetic resonance imaging venography technique to diagnose DVT. This seems to be strange and requires clarification.

Magnetic resonance angiography has a low sensitivity to detect PE and results in a large proportion of inconclusive results (ref: Stein PD, et al. Ann Intern Med 2010;152:434-443; Revel MP, et al. J Thromb Haemost 2012;10:743-750). Therefore, this imaging technique is not yet recommended for diagnosing PE in routine clinical practice according to current guidelines (ref: 2019 ESC Guidelines, Eur Heart J 2020;41:543-603; American Society of Hematology 2018 Guidelines, Blood 2018;2(22):3226-3256).

Lower limb DVT during follow-up will be considered if CUS shows non-compressibility of a venous segment or if an intraluminal filling defect is found on contrast venography. For diagnosis of iliac and caval DVT, magnetic resonance imaging venography will be considered (in addition to duplex ultrasonography and computed tomography), because compression of iliac veins and the inferior vena cava may be technically difficult (see page 12, lines 23-24 and page 13, lines 1-2). Previous studies have shown a high sensitivity of magnetic resonance imaging venography for the diagnosis of iliac DVT (ref: Fraser, DG, et al. Radiology. 2003;226(3)812-20; Fraser DG, Ann Intern Med 2002;136(2)89-98). Only sparse data exist on the diagnosis of caval DVT, but magnetic resonance imaging is among the diagnostic modalities recommended by experts (ref: Alkhouli M, et al. JACC Cardiovasc Interv 2016. 9(7):629-43)

VERSION 2 – REVIEW

REVIEWER	Adam Singer Stony Brook University Medical Center
REVIEW RETURNED	13-Aug-2020

GENERAL COMMENTS	The reviewer completed the checklist but made no further comments.
--

REVIEWER	Hugo Hyung Bok Yoo Botucatu Medical School of São Paulo State University-UNESP
REVIEW RETURNED	17-Aug-2020

GENERAL COMMENTS	Authors adequately responded to my comments
---

REVIEWER	Waldemar Wysokinski Mayo Clinic Rochester, MN, USA
REVIEW RETURNED	26-Aug-2020

GENERAL COMMENTS	1. First, there are unclear criteria for patient recruitment to the study. ISSPE is defined purely based on the anatomical distribution of radiological findings irrespectively of symptomatology and this study recruits asymptomatic as well as symptomatic patients. Since most patients with the diagnosis of asymptomatic ISSPE had CTPA done for other medical problems most often cancer, one can expect that a substantial proportion of patients classified as “low risk” would have symptomatic ISSPE. This group will include those with severe chest pain and shortness of breath related to pleural irritation by peripheral embolic material. Moreover, from the perspective of anatomical involvement by embolic material, this PE category encompasses cases with single
---

isolated subsegmental embolic material as well as multiple, bilateral subsegmental emboli. It will be rather difficult to justify the randomization of very symptomatic patients with multiple, bilateral ISSPE to the placebo group. These patients cannot be excluded based on the “presence of cardiopulmonary compromise” because they do not have a cardiopulmonary compromise. Even, if we assume that severe pleuritic chest pain represents the “cardiopulmonary compromise” – would mild, intermittent, only with a deep breath pain, disqualify the patient? So, what would be the threshold that would still allow study participation? It introduces an important bias to the study.

Thank you for your comment. Our study has clearly defined eligibility criteria for patient recruitment (see inclusion criteria on page 9, lines 7-9, and exclusion criteria in Table 2).

Patients with both symptomatic and incidentally detected, asymptomatic SSPE are eligible for recruitment, because symptoms of PE may be subtle and difficult to elucidate, and because symptomatic and asymptomatic PE appear to have a similar prognosis (see our new reference 35). Specifically, we are unaware of an existing association between pleuritic pain severity (which is somewhat subjective) and prognosis. As we exclude patients with hospital-acquired PE or active cancer (in whom most incidental PEs are found), we expect to enroll only few patients with incidentally detected, asymptomatic SSPE (<2%; see our new reference 35 and Dentali F, et al. *Thromb Res.* 2010; 125(6):518-522). We enroll both patients with single and multiple isolated SSPE, because there is no convincing evidence that these conditions are prognostically different. We have now added this information to page 9, lines 13-17.

In addition, we have clarified that cardiopulmonary compromise relates to hypotension (systolic blood pressure <100mmHg) or hypoxemia (arterial oxygen saturation <92% at ambient air) at the time of presentation (page 9, line 19, and Table 2). Thus, pleuritic pain of any severity, if not associated with hypoxemia or a systolic blood pressure <100 mm Hg, is not a patient selection criterion.

Thank you for the very well documented response. However, in my comment I have not anticipated that symptomatic patient have necessarily poorer prognosis. My concern was related to the original description of current guidelines recommendations (*Chest.* 2016 Feb;149(2):315-52) that represent the basis of this study protocol. This guideline uses symptomatology to direct the providers to use anticoagulation: “Furthermore, a low cardiopulmonary reserve or marked symptoms that cannot be attributed to another condition favor anticoagulant therapy” – so the authors of this study protocol took seriously the first element of this recommendation “a low cardiopulmonary reserve” but ignore the other “marked symptomatology”. In fact, the same guideline also favors anticoagulation over surveillance for calf DVT if patient is very symptomatic: “Severe symptoms favor anticoagulation”. If the authors decide to disregard these recommendations and recruit very symptomatic patients, they have to add to the manuscript that they do it even though current guidelines “favor” anticoagulation in these circumstances. This information also has to be added to the consent form. Participants have to know that one of the study options of management is not favored by current guidelines.

1. The other problem related to patient recruitment could be the radiological accuracy of ISSPE diagnosis. Given the low interobserver agreement ($\kappa = 0.38$) on CTPA interpretation, even

by experienced chest radiologists (17) there could be substantial variability in the accuracy of ISSPE diagnosis at the participating centers. It is rather unfortunate that the central review of CTPA images will be done post-hoc in 6-month batches. Disagreement in the false positive or negative detection of ISSPE could be a problem and may substantially affect the performance of the study particularly that only 276 patients (138 per group) are planned to be recruited. It would be much better if the central review at the Bern University Hospital would be performed within the first few days of patient recruitment.

Please see our answer to comment 3 from reviewer #1.

We agree that interobserver reliability of diagnosing SSPE is suboptimal. As this study aims to reflect real-world practice (where the physician needs to decide on the further management strategy once the SSPE diagnosis is confirmed by the local radiologist/s), the SSPE diagnoses are made according to usual practice at the participating sites. Moreover, although a central review of the CTPA images to confirm the SSPE diagnosis by a team of thoracic radiology experts would be ideal to ensure consistent SSPE diagnoses at all trial sites, this is logistically not feasible due to the short timeline between diagnosis and enrolment, and it would not reflect real-world practice, thus limiting the external validity of our study results. Therefore, the central review of CTPA images will be done post-hoc by 2 blinded thoracic radiologists. Patients in whom SSPE was not confirmed during central review will be excluded from the per-protocol analysis (see page 17, line 6). We have added this explanation to the revised Methods section on page 15, lines 18-21.

The answer to this comment is somewhat contradictory. This protocol claims to be “a multicenter randomized placebo-controlled non-inferiority trial”. Consequently, it should represent standards of randomized study design. Yet, the authors are claiming that their study “aims to reflect real-world practice” – this represents the feature of the opposite spectrum of data sources - “real-world” or “phase IV” evidence such as prospective patient registries, case series and retrospective healthcare or claim database analyses. By the fact that this study uses randomization – it does not “reflect real-world practice” even though both options (surveillance and anticoagulation) are used in clinical practice. Moreover, and more importantly, considering “the low interobserver agreement ($\kappa = 0.38$) on CTPA interpretation, even by experienced chest radiologists” the strategy of “the central review of CTPA images done post-hoc in 6-month batches” – promotes the very possible situation that the patient with more proximal than sub-segmental PE would have US surveillance instead of mandatory anticoagulation. And the patient without PE would be treated with unnecessary anticoagulation. The authors plan to recruit 276 patients (138 per group) – which indicate an average recruitment of 0.76 patient per day if this study would enroll for a year. It is difficult to accent the statement that it would be “logistically difficult” to pursue centralized review of less than one patient per day. .

The calculation of risk and benefit for this study is very narrow. The VTE recurrent rate is estimated by the authors to be 0.8%. The rate of major bleeding, as assessed by EINSTEIN-PE study (18), will be very close to 0.8% too. In this study, the rate of major bleeding in the rivaroxaban arm was 1.1% but only 5.3% of patients were treated for 90 days, 57.3% for 6 and 37.4% for 12 months. The rate of a composite of major and CRNMB was 10.3,

but based on K-M curve analysis about half of bleeding events happened with the first 90 days. Moreover, one should expect bleeding events to be smaller in the group of “low risk” patients eligible for the current study.

Given that we are studying a low-risk population, we agree with the Reviewer that the risk-benefit ratio of clinical surveillance vs. anticoagulation may be relatively small. In the EINSTEIN-PE study, from which patients with an increased bleeding risk were also excluded, the cumulative 90-day incidence of recurrent VTE was around 1.5%, whereas the cumulative 90-day incidence of clinically significant bleeding was about 7%. Still, if our study finds a low 90-day VTE recurrence rate in the clinical surveillance group without anticoagulation (we assumed a 0.8% recurrence based on a previous study of low-risk patients) and a lower bleeding rate than in the anticoagulation group (with is highly probable), it has the potential to improve quality of care and reduce treatment costs in patients with isolated SSPE.

No further comments.

2. Adding “all-cause mortality” to clinically significant bleeding as the safety outcomes introduce the factor which has little to do with study intervention. Low-risk patients have very little risk of death within 90 days and it is unlikely that this event would have anything to do with the rivaroxaban effect but will be rather governed by chance.

We apologize for not having indicated this clearly enough, but clinically significant bleeding and all-cause mortality are two separate secondary outcomes. We have clarified this on page 13, line 3.

No further comments

3. DVT of high potential for PE will not be detected by the current study protocol. This will include DVT of iliac and inferior vena cava thrombosis (25% of DVT) (19,20) of thrombosis of atypical location such as hepatic or gonadal veins, and cerebral venous sinus thrombosis (4% of DVT) (21). Although thromboses of these locations were not included in large randomized clinical trials of acute VTE this study has placebo arm and may put patients with undetected DVT at the substantial risk of lethal PE. The reference cited by the reviewer refers to iliofemoral thrombosis (25% of DVT according to the reviewer’s reference 19). However, isolated iliac or caval DVT is very uncommon, and the latter is mostly associated with the presence of inferior vena cava filters (ref: Jain AK, et al. J Vasc Surg Venous Lymphat Disord. 2018;6(6):724-729; Alkhouli M, et al. JACC Cardiovasc Interv 2016. 9(7):629-43). Similarly, splanchnic vein thromboses and cerebral vein thromboses are uncommon and not typically a source of embolization to the lungs, so it is unlikely that isolated SSPE arises from isolated thromboses at these sites (ref: Ageno W, et al. Blood 2014;124(25):3685-91; Liberman AL, et al. Stroke 2017;48(3):563-567). While guidelines recommend excluding proximal lower extremity DVT if a management strategy without anticoagulation is considered for SSPE, they do not specifically recommend to screen for thromboses at atypical locations (ref: CHEST Guideline and Expert Panel Report. Chest 2016; 149(2):315-352; 2019 ESC Guidelines, Eur Heart J 2020;41:543-603). Thus, our method for detecting DVT is consistent with current guideline recommendations.

The bibliography provided by the authors (Jain AK, et al. J Vasc Surg Venous Lymphat Disord. 2018;6(6):724-729) that presumably should report low prevalence of isolated iliac DVT is not valid because it does not evaluate the prevalence of isolated iliac DVT but the accuracy of US/Doppler study for this particular location. It is not true that isolated iliac DVT is mostly related to IVC filter, it is well known that iliac vein thrombosis is mainly related to May-Turner syndrome. Reference provided by the authors (Alkhouli M, et al. JACC Cardiovasc Interv 2016. 9(7):629-43) represent summary article about inferior vena cava thrombosis without any reference to the incidence of iliac and ilio-caval thrombosis in the cohort of patients with the clinical suspicion of DVT.

Contrary to the authors statement, the isolated pelvic DVT is not uncommon, the early studies reporting that it represents approximately 2% of lower extremities DVT (Br J Radiol 1971; 44: 653–663.; J Vasc Surg 1993; 18:734–741.) were verified by more recent studies using for example MRA (described in details in authors response to comment #4) which revealed that it account for 21.7% of studies in patients with clinical suspicion of DVT (exactly the population of the current study). There are also other studies reporting higher prevalence of isolated iliac DVT (Eur J Vasc Endovasc Surg (2016) 51, 415e420).

I agree that current practice does not call for checking iliac vein thrombosis in every patient with leg DVT, but with the specific situation of the patient already diagnosed with DVT (calf DVT) and the option of using placebo, the problem of the presence of more proximal DVT difficult to diagnosed by US/Doppler – is important – and should be at least included into the limitations of the current protocol.

I agree that splanchnic DVT represent very low risk of PE, but I specifically discussed hepatic and gonadal vein thromboses (not portal, mesenteric, splenic) that have an anatomical potential to cause pulmonary artery embolization. In fact, in the study of 219 patients with ovarian vein thrombosis, PE was identified at presentation in 6%, and another 4 (1.8%) had PE during follow up. (Obstet Gynecol. 2017;130(5):1127-1135). In the group of 154 patients with cerebral venous sinus thrombosis there were 11 patients suffered venous thromboembolism during follow up (2.8/100 patient-years) involving the lower extremity (n=8) or PE (n=5) (Neurology 2006 Sep 12;67(5):814-9).

Again, I believe that current study protocol not including mandatory imaging of pelvic veins such as iliac, inferior vena cava, gonadal or hepatic veins is acceptable but the statement about the risk of missing thrombosis of this location should be added to the limitations of this study.

4. Based on the study description it appears that investigators will not accept magnetic resonance imaging venography to diagnose new PE events for VTE recurrence detection (only CTPA and ventilation-perfusion scan) but would accept magnetic resonance imaging venography technique to diagnose DVT. This seems to be strange and requires clarification. Magnetic resonance angiography has a low sensitivity to detect PE and results in a large proportion of inconclusive results (ref: Stein PD, et al. Ann Intern Med 2010;152:434-443; Revel MP, et al. J Thromb Haemost 2012;10:743-750). Therefore, this imaging technique is not yet recommended for diagnosing PE in routine clinical practice according to current guidelines (ref: 2019 ESC

	Guidelines, Eur Heart J 2020;41:543-603; American Society of Hematology 2018 Guidelines, Blood 2018;2(22):3226-3256). Lower limb DVT during follow-up will be considered if CUS shows non-compressibility of a venous segment or if an intraluminal filling defect is found on contrast venography. For diagnosis of iliac and caval DVT, magnetic resonance imaging venography will be considered (in addition to duplex ultrasonography and computed tomography), because compression of iliac veins and the inferior vena cava may be technically difficult (see page 12, lines 23-24 and page 13, lines 1-2). Previous studies have shown a high sensitivity of magnetic resonance imaging venography for the diagnosis of iliac DVT (ref: Fraser, DG, et al. Radiology. 2003;226(3):812-20; Fraser DG, Ann Intern Med 2002;136(2):89-98). Only sparse data exist on the diagnosis of caval DVT, but magnetic resonance imaging is among the diagnostic modalities recommended by experts (ref: Alkhouli M, et al. JACC Cardiovasc Interv 2016. 9(7):629-43). I agree that using MRA as a first line of diagnostic tool for PE is not recommended. The overall sensitivity of MRA for PE is only 57% (PIOPED III). However, the main reason for low sensitivity is relatively high proportion of patients who would have technically inadequate examination (MRA requires long acquisition times and a lot of dyspneic patients cannot hold their breath). Excluding patients with technically inadequate studies increased the sensitivity to 78 which is comparable to CT-angiogram. The PIOPED II study demonstrated an overall sensitivity of multidetector CT angiography of 83%. However, CT-angiogram negative predictive value highly depends on pre-test clinical probability. It drops to 60% if the pre-test probability is high. Also positive predictive value is only 58% in patients with a low pre-test likelihood of PE. I believe that the authors may consider using MRA angiography in the situation of minimal respiratory compromise and allergy to iodine, or in young patients in whom ionizing radiation can be avoided. I do not question utility of MRA for DVT diagnosis although CT-angiography is more often use for this indication.
--	---

VERSION 2 – AUTHOR RESPONSE

Response from Reviewer 3 to our answers to his comments from revision #1

1. Thank you for the very well documented response. However, in my comment I have not anticipated that symptomatic patient have necessarily poorer prognosis. My concern was related to the original description of current guidelines recommendations (Chest. 2016 Feb;149(2):315-52) that represent the basis of this study protocol. This guideline uses symptomatology to direct the providers to use anticoagulation: “Furthermore, a low cardiopulmonary reserve or marked symptoms that cannot be attributed to another condition favor anticoagulant therapy” – so the authors of this study protocol took seriously the first element of this recommendation “a low cardiopulmonary reserve” but ignore the other “marked symptomatology”. In fact, the same guideline also favors anticoagulation over surveillance for calf DVT if patient is very symptomatic: “Severe symptoms favor anticoagulation”. If the authors decide to disregard these recommendations and recruit very symptomatic patients, they have to add to the manuscript that they do it even though current guidelines “favor” anticoagulation in

these circumstances. This information also has to be added to the consent form. Participants have to know that one of the study options of management is not favored by current guidelines.

We acknowledge the reviewer's comment concerning the guideline recommendations. However, we are not aware of any convincing evidence demonstrating that symptomatic SSPE has a higher risk of recurrence or clinical deterioration compared to asymptomatic SSPE. Also, to our knowledge, there is no evidence that the severity of PE-related symptoms, such as chest pain, is associated with prognosis. We would presume that patients who have marked dyspnea are hypoxemic, in which case they would be excluded from study participation. Anticoagulation treatment in PE reduces the risk of clot progression and VTE recurrence, but does not primarily serve to treat symptoms. Thus, in our view, the decision on whether or not to anticoagulate SSPE should be based on validated prognostic factors rather than symptoms. In fact, an ongoing Canadian prospective management study investigating a treatment strategy without anticoagulation in SSPE only includes symptomatic patients (ClinicalTrials.gov identifier NCT01455818). The protocol of our study has been developed based on the experience and input of Prof. Carrier, who is not only the PI of the ongoing prospective management study, but also a steering committee member of our study. No relevant safety concerns have arisen in the ongoing prospective management study given that recruitment is still ongoing after >250 patients have already been enrolled (personal communication Prof. Carrier). In addition, guidelines differ in their recommendations concerning the exact criteria for selecting SSPE patients in whom anticoagulation treatment could be withheld, reflecting the low quality of evidence. For example, the 2019 ESC guidelines for PE do not recommend considering symptoms in the decision whether or not to anticoagulate SSPE (ref: Konstantinides, et al. 2019 ESC Guidelines for the diagnosis and management of acute pulmonary embolism developed in collaboration with the European Respiratory Society (ERS). Eur Heart J 2020). Finally, our protocol has undergone a very thorough review by an international expert panel and by Swiss Ethics committees, which did not find any safety issues regarding the inclusion of symptomatic patients. The ethics committees have also approved the content of our current patient information and consent forms. Thus, we hope that the Reviewer agrees with our decision not to modify the research protocol and the patient consent form.

2. The answer to this comment is somewhat contradictory. This protocol claims to be "a multicenter randomized placebo-controlled non-inferiority trial". Consequently, it should represent standards of randomized study design. Yet, the authors are claiming that their study "aims to reflect real-world practice" – this represents the feature of the opposite spectrum of data sources - "real-world" or "phase IV" evidence such as prospective patient registries, case series and retrospective healthcare or claim database analyses. By the fact that this study uses randomization – it does not "reflect real-world practice" even though both options (surveillance and anticoagulation) are used in clinical practice. Moreover, and more importantly, considering "the low interobserver agreement ($\kappa = 0.38$) on CTPA interpretation, even by experienced chest radiologists" the strategy of "the central review of CTPA images done post-hoc in 6-month batches" – promotes the very possible situation that the patient with more proximal than sub-segmental PE would have US surveillance instead of mandatory anticoagulation. And the patient without PE would be treated with unnecessary anticoagulation. The authors plan to recruit 276 patients (138 per group) – which indicate an average recruitment of 0.76 patient per day if this study would enroll for a year. It is difficult to accent the statement that it would be "logistically difficult" to pursue centralized review of less than one patient per day.

Thank you for the comment. Of course, a RCT with its distinct eligibility criteria, standardized interventions and outcomes assessment can never fully reflect "real world" practice. However, we believe that some aspects of study, i.e., the inclusion of patients with SSPE based on the final decision of the local radiologists, are pragmatic and reflect what is actually done in every-day clinical practice, where the diagnosis of SSPE is not based on a panel decision of experienced thoracic radiologists but on the decision of the local radiologist in charge. An immediate central review by a panel of thoracic radiologists would not only drastically reduce the generalizability of our study, it is

logistically not feasible. This not a matter of SSPE volume, but of the narrow time window between SSPE diagnosis and possible enrolment. Given that this is an international multicenter trial with a 24/7 recruitment, immediate transfer of all CTPA images from SSPE patients recruited at any given study site to undergo central review by thoracic radiology experts in Bern within a few hours is logistically not realistic, especially, since strict requirements to guarantee confidential electronic image transfer must be met. We are aware that our steering committee's decision to base the diagnosis of SSPE on a local CT review carries a risk of false positive and false negative results, but this risk is inherent to all imaging and is not higher in our study than in current clinical practice. In light of these arguments, the retrospective central CTPA-image review was deemed acceptable by the international reviewer panel and Swiss ethics committees.

3. The bibliography provided by the authors (Jain AK, et al. *J Vasc Surg Venous Lymphat Disord.* 2018;6(6):724-729) that presumably should report low prevalence of isolated iliac DVT is not valid because it does not evaluate the prevalence of isolated iliac DVT but the accuracy of US/Doppler study for this particular location. It is not true that isolated iliac DVT is mostly related to IVC filter, it is well known that iliac vein thrombosis is mainly related to May-Turner syndrome. Reference provided by the authors (Alkhouli M, et al. *JACC Cardiovasc Interv* 2016. 9(7):629-43) represent summary article about inferior vena cava thrombosis without any reference to the incidence of iliac and ilio-caval thrombosis in the cohort of patients with the clinical suspicion of DVT.

Contrary to the authors statement, the isolated pelvic DVT is not uncommon, the early studies reporting that it represents approximately 2% of lower extremities DVT (*Br J Radiol* 1971; 44: 653–663.; *J Vasc Surg* 1993; 18:734–741.) were verified by more recent studies using for example MRA (described in details in authors response to comment #4) which revealed that it account for 21.7% of studies in patients with clinical suspicion of DVT (exactly the population of the current study). There are also other studies reporting higher prevalence of isolated iliac DVT (*Eur J Vasc Endovasc Surg* (2016) 51, 415e420).

I agree that current practice does not call for checking iliac vein thrombosis in every patient with leg DVT, but with the specific situation of the patient already diagnosed with DVT (calf DVT) and the option of using placebo, the problem of the presence of more proximal DVT difficult to diagnosed by US/Doppler – is important – and should be at least included into the limitations of the current protocol. I agree that splanchnic DVT represent very low risk of PE, but I specifically discussed hepatic and gonadal vein thromboses (not portal, mesenteric, splenic) that have an anatomical potential to cause pulmonary artery embolization. In fact, in the study of 219 patients with ovarian vein thrombosis, PE was identified at presentation in 6%, and another 4 (1.8%) had PE during follow up. (*Obstet Gynecol.* 2017;130(5):1127-1135). In the group of 154 patients with cerebral venous sinus thrombosis there were 11 patients suffered venous thromboembolism during follow up (2.8/100 patient-years) involving the lower extremity (n=8) or PE (n=5) (*Neurology* 2006 Sep 12;67(5):814-9).

Again, I believe that current study protocol not including mandatory imaging of pelvic veins such as iliac, inferior vena cava, gonadal or hepatic veins is acceptable but the statement about the risk of missing thrombosis of this location should be added to the limitations of this study.

As the Reviewer correctly points out, there is a risk that a patient with SSPE may have a concomitant (asymptomatic) DVT that is not captured by bilateral leg vein compression ultrasonography. However, even if a small proportion of patients with thromboses at atypical locations (including cerebral, splanchnic, or ovarian vein thrombosis) develop PE as demonstrated in the above cited studies, these are rare thrombotic conditions that often occur in specific clinical situations representing study exclusion criteria (cancer, pregnancy, puerperium; ref: McBane R, Tafur A, Wysokinski WE. Acquired and congenital risk factors associated with cerebral venous sinus thrombosis. *Thromb Res* 2010; Lenz CJ, Wysokinski WE, et al. Ovarian Vein Thrombosis: Incidence of Recurrent Venous Thromboembolism and Survival. *Obstet Gynecol* 2017; Mimier MK, et al. Thrombosis of atypical location: how to treat patients in the era of direct oral anticoagulants? *Pol Arch Intern Med* 2018). The risk that such patients would present with an isolated SSPE is even lower. We agree that whole-leg

compression ultrasonography (CUS) is not an effective tool to diagnose iliac vein thrombosis. While the prevalence of isolated iliac vein DVT varies widely depending on the type of imaging used, a meta-analysis of randomized trials and prospective management studies has convincingly shown that in patients with suspected DVT in whom DVT has been excluded by a single whole-leg CUS and in whom anticoagulation is withheld, the risk of recurrent VTE is very low (ref: Johnson SA, et al. Risk of deep vein thrombosis following a single negative whole-leg compression ultrasound: a systematic review and meta-analysis. JAMA 2010). Although we acknowledge that the studies from this meta-analysis do not directly relate to patients with isolated SSPE, they provide reassurance that the rate of isolated iliac vein thromboses in patients even with symptoms or signs of DVT must be very low. Besides, patients with distal DVT are excluded from study participation. To address the Reviewer's concern, we added the following sentence to page 9, lines 25-26 and page 10, lines 1-8 of the revised manuscript:

„ We cannot fully exclude the small possibility that an eligible patient with SSPE in whom the presence of a leg vein DVT was ruled out by a bilateral whole-leg CUS may have a concomitant isolated iliac vein thrombosis or a thrombosis at an unusual site (e.g., in the cerebral, splanchnic, or ovarian veins). However, these are rare thrombotic conditions that often occur in specific clinical situations representing study exclusion criteria (cancer, pregnancy, puerperium).⁴⁸ The risk that such patients would present with an isolated SSPE is even lower. Although whole-leg CUS is not an effective method to exclude an isolated iliac vein thrombosis, a meta-analysis of randomized trials and prospective management studies has convincingly shown that in patients with suspected DVT in whom DVT has been excluded by a single whole-leg CUS and in whom anticoagulation is withheld, the risk of recurrent VTE is very low.⁴⁹ ”

4. I agree that using MRA as a first line of diagnostic tool for PE is not recommended. The overall sensitivity of MRA for PE is only 57% (PIOPED III). However, the main reason for low sensitivity is relatively high proportion of patients who would have technically inadequate examination (MRA requires long acquisition times and a lot of dyspneic patients cannot hold their breath). Excluding patients with technically inadequate studies increased the sensitivity to 78 which is comparable to CT-angiogram. The PIOPED II study demonstrated an overall sensitivity of multidetector CT angiography of 83%. However, CT-angiogram negative predictive value highly depends on pre-test clinical probability. It drops to 60% if the pre-test probability is high. Also positive predictive value is only 58% in patients with a low pre-test likelihood of PE.

I believe that the authors may consider using MRA angiography in the situation of minimal respiratory compromise and allergy to iodine, or in young patients in whom ionizing radiation can be avoided. I do not question utility of MRA for DVT diagnosis although CT-angiography is more often use for this indication.

Thank you for your comment. MRA is not routinely used as a diagnostic tool for PE in any of our study sites, even in young patients. Moreover, for the sake of comparability, we prefer using a single imaging technique (CTPA) for diagnosing SSPE rather than different methods with potentially differing test characteristics. We therefore hope that the Reviewer understands our decision to use CTPA only.